# A Polyvalent Adhesin–Toxoid Multiepitope-Fusion-Antigen-Induced Functional Antibodies against Five Enterotoxigenic *Escherichia coli* Adhesins (CS7, CS12, CS14, CS17, and CS21) but Not Enterotoxins (LT and STa)

**DOI:** 10.3390/microorganisms11102473

**Published:** 2023-10-01

**Authors:** Siqi Li, Hyesuk Seo, Ipshita Upadhyay, Weiping Zhang

**Affiliations:** Department of Pathobiology, University of Illinois at Urbana-Champaign, Urbana, IL 61802, USA

**Keywords:** ETEC (enterotoxigenic *Escherichia coli*), adhesin, vaccine, diarrhea, polyvalent protein, MEFA (multiepitope fusion antigen)

## Abstract

The increasing prevalence and association with moderate-to-severe diarrhea make enterotoxigenic *Escherichia coli* (ETEC) adhesins CS7, CS12, CS14, CS17, and CS21 potential targets of ETEC vaccines. Currently, there are no vaccines licensed to protect against ETEC, a top cause of children’s diarrhea and travelers’ diarrhea. Recently, a polyvalent adhesin protein (adhesin MEFA-II) was demonstrated to induce antibodies that inhibited adherence from these five ETEC adhesins and reduced the enterotoxicity of ETEC heat-stable toxin (STa), which plays a key role in causing ETEC-associated diarrhea. To improve adhesin MEFA-II for functional antibodies against STa toxin and the other ETEC toxin, heat-labile toxin (LT), we modified adhesin MEFA-II by adding another STa toxoid and an LT epitope; we examined the new antigen immunogenicity (to five adhesins and two toxins) and more importantly antibody functions against ETEC adherence and STa and LT enterotoxicity. Data show that mice intramuscularly immunized with the new antigen (adhesin MEFA-IIb) developed robust IgG responses to the targeted adhesins (CS7, CS12, CS14, CS17, and CS21) and toxins (STa and LT). Mouse antibodies inhibited the adherence of ETEC strains expressing any of these five adhesins but failed to neutralize STa or LT enterotoxicity. In further studies, rabbits intramuscularly immunized with adhesin MEFA-IIb developed robust antigen-specific antibodies; when challenged with an ETEC isolate expressing CS21 adhesin (JF2101, CS21, and STa), the immunized rabbits showed a significant reduction in intestinal colonization by ETEC bacteria. These data indicate that adhesin MEFA-IIb is broadly immunogenic and induces functional antibodies against the targeted ETEC adhesins but not the toxins.

## 1. Introduction

*Escherichia coli* strains that produce enterotoxins, enterotoxigenic *E. coli* (ETEC), is one of the top four causes of diarrhea in children younger than five years in low-to-middle-income countries (children’s diarrhea) and the most common cause of diarrhea in travelers who travel from industrialized countries to ETEC endemic countries or regions (travelers’ diarrhea) [1,2,3]. ETEC strains are estimated to cause hundreds of millions of diarrhea clinical cases and up to nearly a hundred thousand deaths annually. Currently, we do not have -effective countermeasures against ETEC-associated diarrhea. Clean drinking water and improved sanitation systems and personal hygiene (WASH—water, sanitation, and hygiene) can effectively prevent ETEC and other enteric infections; however, the WASH program is unlikely to be implemented soon for many resource-limited countries or communities due to financial restraints. Antibiotic drugs that were used to treat severe diarrhea cases become ineffective since ETEC strains, like many other enteric bacteria, are acquiring resistance to antibiotics at an increasingly alarming rate. Vaccines are regarded as a practical and more cost-effective countermeasure against ETEC infection. Unfortunately, we do not have vaccines currently licensed for ETEC diarrhea [4].

Immunologic heterogeneity becomes a key challenge in ETEC vaccine development since different ETEC strains or pathovars produce different virulence factors [4,5,6]. Moreover, the prevalence of ETEC strains to cause diarrhea can shift geographically and chronically [7,8]. There are two major types of ETEC virulence determinants: adhesins and enterotoxins. ETEC enterotoxins, namely heat-labile toxin (LT) and heat-stable toxin (STa), elevate intracellular cyclic adenosine monophosphate (AMP) or guanosine monophosphate (GMP) levels in host small intestinal epithelial cells and stimulate water and fluid hypersecretion and watery diarrhea. Adhesins mediate the attachment of ETEC bacteria to host cell receptors and subsequent colonization in small intestines. While more than 25 adhesins, including colonization factor antigens (CFAs) and coli surface antigens (CSs), have been identified from the ETEC strains isolated from diarrhea patients, the strains expressing any of the seven adhesins, CFA/I, CFA/II (CS1, CS2, and CS3), and CFA/IV (CS4, SC5, and CS6), along with enterotoxic heat-stable toxin (STa) and/or heat-labile toxin (LT), are identified as the dominant sources associated with ETEC-associated diarrhea [7,9,10,11], and ETEC strains expressing these seven adhesins tend to be more virulent to cause moderate-to-severe clinical cases [12]. Therefore, CFA/I and CS1 to CS6 adhesins are historically targeted for ETEC vaccine development [6].

ETEC strains expressing these seven adhesins have recently been estimated to be responsible for about two-thirds of ETEC-associated clinical cases, and other strains producing different adhesins (other than CFA/I and CS1–CS6) are also found to play a major role in causing children’s or travelers’ diarrhea [12]. Systematic survey and cohort case studies indicated that adhesins CS21, CS14, CS7, CS12, and CS17 are prevalent among ETEC strains isolated from diarrheal patients and associated with moderate-to-severe diarrhea [7,11,12,13,14,15]. Therefore, these five adhesins are also considered to be the targets for ETEC vaccine development. If a vaccine product that protects the seven most prevalent adhesins (CFA/I and CS1–CS6) could expand the coverage to five additional adhesins (CS7, CS12, CS14, CS17, and CS21), this vaccine would improve protection against STa-only ETEC infection from 64.3% to 86.2%, against STa-positive ETEC (STa alone or together with LT) strains from 66% to 80.6%, and against LT-only ETEC from 25% to 47.3% [12].

By applying an epitope- and structure-based multiepitope fusion antigen (MEFA) vaccinology platform [16], we initially constructed a polyvalent protein immunogen to induce cross-protective immunity against these five ETEC adhesins (CS7, CS12, CS14, CS17, and CS21) and found that this new MEFA protein antigen (termed as adhesin MEFA-II, to be differentiated with CFA/I/II/IV MEFA) is broadly immunogenic and induces functional antibodies against the five targeted ETEC adhesins as well as ETEC STa toxin [17]. Adhesin MEFA-II used adhesin CFA/I major subunit CfaB as the backbone to present epitopes from the major subunits of these five adhesins (CsvA of CS7, CswA of CS12, CsuA of CS14, CsbA of CS17, and LngA of CS21) as well as an STa toxoid (STa_N12S_), as a polyvalent protein immunogen. Mice intramuscularly immunized with adhesin MEFA-II developed robust IgG responses to the five targeted adhesins as well as STa toxin. Moreover, the induced antibodies inhibited the adherence from the five adhesins and reduced STa enterotoxicity [17].

If adhesin MEFA-II could be further improved to enhance functional anti-STa immunity and also induce neutralizing antibodies against the other ETEC toxin (LT), this protein antigen can be potentially combined with MEFA CFA/I/II/IV, another polyvalent adhesin antigen previously constructed to target seven different ETEC adhesins, for an ETEC vaccine candidate. CFA/I/II/IV MEFA protein was demonstrated to induce functional antibodies against any of the seven most important ETEC adhesins (CFA/I and CS1-CS6) but also prevented intestinal colonization with ETEC bacteria [18,19,20,21]. Such a vaccine would synergistically protect against twelve ETEC adhesins and both toxins. The vaccine-induced anti-adhesin antibodies and antitoxin antibodies would synergistically protect against ETEC strains producing any of the twelve adhesins and either toxin and also individually (by antitoxin antibodies) against the remaining ETEC strains expressing the other adhesins but one or both ETEC toxins, thus potentially achieving a truly effective vaccine against ETEC-associated children’s diarrhea and travelers’ diarrhea.

In this study, we modified the adhesin MEFA-II protein immunogen to carry another copy of STa toxoid STa_N12S_ to enhance anti-STa immunity and also an epitope of LT toxin to induce anti-LT antibodies. We then immunized mice with the modified adhesin MEFA protein to assess broad anti-adhesin and antitoxin responses and antibody in vitro protection against ETEC bacteria adherence and toxin enterotoxicity. Moreover, we applied a rabbit colonization model by immunizing rabbits with the protein antigen and challenging the immunized rabbits with an ETEC strain to evaluate in vivo protection against ETEC bacteria intestinal colonization.

## 2. Materials and Methods

### 2.1. Ethics Statement

Mice and rabbits used in the immunization or challenge studies complied with the policy and guidelines of the Animal Welfare Act (1996 National Council Research Guidelines), the Policy on Humane Care and Use of Laboratory Animals (Public Health Service), and the United States Department of Agriculture Animal Welfare Act Regulations. Research protocols with animals were reviewed and approved by the Institutional Animal Care and Use Committee at the University of Illinois at Urbana-Champaign (mouse protocol #18260, rabbit protocol #20099). Animal care and use were supervised by institutional attending veterinarians and staff.

### 2.2. Bacteria Strains Used in the Study

Vector pET28a (Novagen; Madison, WI, USA) and *E. coli* BL21 CodonPlus (DE3) strain (Agilent Technologies; Santa Clara, CA, USA) were used to clone and express the modified adhesin protein named MEFA-IIb (9578) as the immunogen for mouse or rabbit immunization. The vector and the *E. coli* strain were also used to produce CS7 major subunit protein CsvA (9611), CS12 major subunit CswA (9604), CS14 major subunit CsuA (9579), CS17 major subunit CsbA (9580), or CS21 major subunit LngA (9582) as the ELISA coating antigens for antigen-specific antibody titration. ETEC field isolates, including JF2327 (CS7 and LT), JF3276 (CS12/CS20, LT, and STa), JF2125 (CS14 and LT), JF2350 (CS17 and LT), and JF2101 (CS21 and STa) [22], were used for antibody adherence inhibition assays or rabbit challenge study.

### 2.3. Adhesin MEFA-IIb Construction and Characterization

Adhesin MEFA-IIb was initially constructed in silico by selecting the strongly immunogenic and structure-stable adhesin CFA/I major subunit CfaB as the backbone to present two copies of STa toxoid STa_N12S_, an epitope from the A subunit of LT toxin, and an epitope from the major subunit of each of the five targeted adhesin major subunit. Therefore, eight continuous B-cell immunodominant epitopes from the CfaB backbone, which were predicted with the B-cell epitope prediction program BepiPred-2.0 (http://tools.iedb.org/bcell/help/#Bepipred-2.0 accessed on 26 September 2023), were substituted with two STa toxoids, an LT_A_ epitope, and five adhesin epitopes. The resultant multiepitope fusion antigen was examined for epitope antigenicity, surface exposure, and epitope proposition with the IEBD program (www.iebd.org accessed on 26 September 2023), PyMol (www.pymol.org accessed on 26 September 2023), and Phyre2 (www.sbg.bio.ic.ac.uk/~phyre2 accessed on 26 September 2023), as we described previously [16,17]. The protein structure stability of the resultant adhesin MEFA was estimated with ExPASy (www.expasy.org accessed on 26 September 2023).

After epitope antigenicity, epitope surface exposure, and protein structural stability were confirmed in silico, the adhesin MEFA-IIb gene was codon-optimized and synthesized using GenScript Biotech (Piscataway, NJ, USA). The synthetic gene was subsequently cloned into vector pET28a at the NcoI and EagI sites. The cloned adhesin MEFA-IIb gene was expressed by *E. coli* BL21 CodonPlus (DE3).

Adhesin MEFA-IIb protein was expressed in Yeast Extract Tryptone (YT; Fisher Scientific) with kanamycin (30 μg/mL) and inducer isopropyl β-D-1-thiogalactopyranoside (IPTG; Sigma, 1 mM); inclusion body proteins were extracted using bacterial protein extraction reagent (B-PER; Thermo Fisher Scientific, Waltham, MA, USA) by following the manufacturer’s protocol. As described previously [17,19,23], adhesin MEFA-IIb inclusion body protein was then solubilized with 50 mM CAPs buffer (pH 11.0) supplemented with 0.3% N-lauroylsarcosine and 1 mM dithiothreitol (DTT). Solubilized adhesin MEFA-IIb protein was subsequently refolded and dialyzed with 1 M Tris-HCl (pH 8.5) supplemented with 0.1 mM DTT and followed with 1 M Tris-HCl buffer without DTT. Refolded protein was measured for protein concentration with the Lowery Method, adjusted to 1 mg/mL with PBS, aliquoted, and stored at −80 °C.

Protein was examined in 15% sodium dodecyl sulfate polyacrylamide gel electrophoresis (SDS-PAGE), visualized with Coomassie blue staining to assess protein purity, and characterized in Western blot with anti-CfaB backbone (9472 backbone) mouse polyclonal antibodies (1:5000).

### 2.4. Mouse Immunization

Eight-week-old female BALB/c mice (Charles River Laboratories, Wilmington, MA, USA), eight per group, were intramuscularly immunized with 25 μg adhesin MEFA-IIb protein (in 25 μL) adjuvanted with 0.1 μg double mutant LT (dmLT, LT_R192g/L211A_) or 25 μL PBS. Mice received two booster injections (the same dose as the primary) at a two-week interval. Mice were sacrificed in two weeks followed by the second booster. Mouse serum samples were collected prior to the primary and two weeks after the 2nd booster and stored at −80 °C until use.

Additionally, fecal pellets were collected two weeks after the final booster and at necropsy. Fecal pellets were suspended in fecal reconstitution buffer (10 mM Tris, 100 mM NaCl, 0.05% Tween-20, 5 mM sodium azide; pH 7.4) supplemented with 0.2 mg/mL phenylmethylsulfonyl fluoride, at the ratio of 1 g in 5 mL. The fecal suspension was centrifuged, and supernatants were collected and stored at −80 °C.

### 2.5. Mouse Antigen-Specific Antibody Titration

Mouse serum samples collected before the primary and two weeks after the final booster and fecal suspension samples collected at necropsy were titrated for IgG and IgA antibodies to the five ETEC adhesins and two toxins in ELISAs, as described previously [19,24,25]. Briefly, 2HB 96-well microtiter plates (Thermo Fisher Scientific) coated with (100 ng per well) recombinant major subunit protein of each of the five targeted adhesins (CS7, CS12, CS14, CS17, and CS21) or cholera toxin (CT; Sigma, St. Louis, MO, USA) were used to titrate antibody responses to the five adhesins and LT toxin; Corning CoStar 96-well plates (Fisher Scientific) coated with STa–ovalbumin conjugates, 10 ng per well, were used to titrate anti-STa antibodies. Two-fold serum dilutions (1:200 to 1:125,600) or fecal suspension dilutions (1:10 to 1:640) were added to the wells with coated antigens and incubated at 37 °C for 1h. After washes with PBST, wells were incubated with horseradish peroxidase (HRP)-conjugated goat anti-mouse IgG or IgA (1:5000; Bethyl Laboratories, Montgomery, TX, USA), followed by washes with PBST and incubation with 3,3′,5,5′-tetramethylbenzidine (TMB) microwell peroxidase substrate system 2C (Thermo Fisher Scientific). OD_650_ readings were calculated to antibody titers, by multiplying the highest serum or fecal dilution that gave an OD over 0.3 (after subtraction with background OD) with the adjusted OD, and presented in log_10_ scale.2.6. Mouse Serum Antibody Adherence Inhibition against ETEC Strains Expressing the Five Targeted Adhesins and Neutralization against STa and LT Enterotoxicity

Mouse serum samples from the immunized group or the control group were examined for in vitro antibody-induced protection against ETEC adherence and STa or LT enterotoxicity. As we described previously [17,19,25], 3.0 to 3.5 × 10^6^ CFUs of the ETEC strain that expresses one of the five targeted adhesins, after treatment with 10% mannose, were mixed with 15 μL heat-inactivated serum sample and incubated at room temperature for 1 h (50 rpm), and then transferred to one of the wells of a 24-well tissue culture plate containing 95–100% confluent monolayered Caco-2 cells (7 × 10^5^ cells; ATCC, HTB-37). After 1 h incubation in a 37 °C 5% CO_2_ incubator, cells were rinsed with PBS to remove non-adherent ETEC bacteria, dislodged with Triton (0.5%; Sigma), collected, tenfold diluted, and plated on LB agar plates. Overnight-grown CFUs were counted and presented in percentage, with 100% referred to the CFUs from the treatment with the control mouse sera.

Antibody neutralization activity against STa and LT enterotoxicity from the mouse serum sample was examined in T-84 cells (ATCC, CCL-248), using a cyclic AMP or GMP EIA kit (Enzo Life Sciences, Farmingdale, NY, USA). Briefly, as previously described [17,19,25], 10 ng CT or 2 ng STa toxin was incubated with 30 μL heat-inactivated mouse sera of the immunized group or the control group for 30 min at room temperature. The toxin–serum mixture was transferred to a 24-well tissue culture plate well containing 95–100% confluent monolayered T-84 cells and incubated in a 37 °C 5% CO_2_ incubator for 1 h (for STa, cGMP) or 3 h (for CT, cAMP). After three washes with PBS to remove extracellular cGMP or cAMP, cells were dislodged and lysed to release intracellular cGMP or cAMP. Cell lysates were collected and measured for intracellular cGMP or cAMP concentrations (nM, picomole per mL), by following the manufacturer’s protocol.

### 2.6. Rabbit Immunization and Challenge

Adult New Zealand White (NZW) rabbits (Charles River Laboratories), four per group, were intramuscularly immunized with 200 μg adhesin MEFA-IIb protein or PBS (in 200 μL), adjuvanted with 1 μg dmLT (1 μg/uL). Rabbits received two boosters at an interval of two weeks. Rabbit serum and fecal pellet samples collected before the primary and two weeks after the second booster were used to examine antigen-specific antibody titration as well as antibody functional activities against ETEC bacteria adherence or toxin enterotoxicity, as described above, except for the use of HRP-conjugated goat anti-rabbit IgG or IgA (Bethyl Laboratories) as the secondary antibodies.

Two weeks after the second booster, all rabbits were orogastrically challenged with ETEC field isolate JF2101 (CS21 and STa) to assess adhesin MEFA-IIb in vivo protection against ETEC intestinal colonization. As we described previously [21,23], rabbits were first administered with famotidine (0.75 mg per kg body weight) given intravenously (IV) at the ear vein 3 h before inoculation. Just before inoculation, rabbits were sedated with dexmedetomidine (0.1 mg per kg body weight) given intramuscularly and then anesthetized with the inhalation of 5% isoflurane, followed by inoculation with 3 mL 5% sodium bicarbonate, 5 × 10^10^ CFU ETEC strain JF2101 (in 5 mL), and finally 3 mL 5% sodium bicarbonate.

Challenged rabbits were closely monitored for abnormal signs, including vomiting, lethargy, or loose fecal output. Twenty-four hours post-inoculation, all rabbits were sedated with dexmedetomidine, anesthetized with the inhalation of isoflurane, euthanized with exsanguination and KCl (2 mg/mL) intracardiac injection, and necropsied. At necropsy, intestines were visually examined for fluid accumulation, and a length of 10 cm distal ileal segment was collected in sterile bags and transported to the laboratory for bacteria quantitated colonization study. Briefly, each ileal segment was opened longitudinally, rinsed with sterile PBS to remove feces, and ground in PBS (1 g ileal sample in 9 mL PBS) with a glass tissue grinder. The ground sample was then serially diluted and plated on LB agar plates overnight at 37 °C. CFUs were counted and recorded.

Additionally, cecal contents were collected from an individual rabbit at necropsy, suspended in fecal reconstitution buffer, and suspension supernatants were collected.

### 2.7. Statistical Analysis

Data are presented in means and standard deviations, with antibody titers (in triplicates) in log_10_ (for serum IgG) or log_2_ (for rabbit cecum content IgA), antibody adherence inhibition in percentage (in triplicates), antibody toxin neutralization in nM (in duplicates), and ETEC bacteria intestinal colonization in CFUs (in triplicates). Differences between the immunized group and the control group were analyzed with one-way ANOVA with Turkey’s test, by using GraphPad Prism 5 (GraphPad, San Diego, CA, USA). A *p*-value of less than 0.05 indicates a significant difference.

## 3. Results

### 3.1. Adhesin MEFA-IIb Presenting Epitopes of the Five ETEC Adhesins (CS7, CS12, CS14, CS17, and CS21), Two STa Toxoids, and an LT Epitope Was Constructed and Expressed

Epitopes “AGSPVTRSDTTS”, “GSNGQANNNDASQ”, “TSGTAPSAGKYQ”, “ADTQGTAPEAGNY”, and “VAMKDAYQRDGKYPDF” were identified in silico as the continuous immunodominant B-cell epitopes from the major subunits of adhesins CS7 (CsvA), CS12 (CswA), CS14 (CsuA), CS17 (CsbA), and CS21 (LngA), respectively. These five ETEC adhesin epitopes, two copies of full-length STa_N12S_ toxoid (MNSSNYCCELCCSPACTGCY) and an epitope from LT A subunit (SPHPYEQEVSALTA) were presented on the backbone protein CfaB, the major subunit of CFA/I adhesin (Figure 1A,B), by replacing backbone epitopes, with the assistance of the MEFA vaccinology platform [16]. The resultant MEFA-IIb protein was confirmed in silico to be structurally stable (with an instability index of 39), with each adhesin and LT epitope antigenic (note: STa and STa toxoids are poorly immunogenic) and surface-exposed. This MEFA-IIb gene was then codon-optimized and synthesized, cloned into expression vector pET28a, and expressed in *E. coli* strain BL21 (DE3). The MEFA-IIb protein had a molecule size of 21.4 kDa; the extracted protein (at 40 mg per liter of culture broth) showed a visually estimated purity of about 95%, based on SDS-PAGE with Coomassie blue staining and was recognized by anti-CS21 polyclonal antibodies in Western blot (Figure 1C).

### 3.2. Mice Immunized with Adhesin MEFA-IIb Developed Strong IgG Titers to Five ETEC Adhesins (CS7, CS12, CS14, CS17, and CS21) as well as to Both ETEC Toxins (LT and STa)

Mice immunized with adhesin MEFA-IIb protein developed high IgG titers to adhesins CS7, CS12, CS14, CS17, and CS21, as well as to both LT and STa toxins (Figure 2). Anti-CS7, -CS12, -CS14, -CS17, and anti-CS21 titers (log_10_) were detected at 3.1 ± 0.23, 3.3 ± 0.22, 3.7 ± 0.29, 2.9 ± 0.52, and 4.2 ± 0.44, respectively, in the sera of mice IM immunized with the MEFA-IIb protein. Anti-LT and anti-STa serum IgG titers were detected at 1.8 ± 0.17 and 2.6 ± 0.36 (log_10_). No antigen-specific antibodies were detected from the serum or fecal samples of the control mice or the serum or fecal samples collected before the primary dose. No antigen-specific IgA responses were detected from mouse serum or fecal samples.

### 3.3. Adhesin MEFA-IIb-Induced Mouse Serum Antibodies Inhibited the Adherence of CS7, CS12, CS14, CS17, and CS21 Adhesins

ETEC isolates expressing adhesin CS7 (JF2327), CS12 (JF3276), CS14 (JF2125), CS17 (JF2350), or CS21 (JF2101) incubated with heat-inactivated sera from the immunized mice significantly reduced bacteria adherence to Caco-2 cells, when compared to adherence from the same bacteria that were incubated with the control mouse sera (Figure 3). MEFA-IIb-induced antibodies inhibited the adherence of the CS7, CS12, CS14, CS17, or CS21 ETEC isolate by 53%, 51%, 64%, 49%, and 58%, respectively.

### 3.4. Adhesin MEFA-IIb-Induced Antibodies Did Not Neutralize STa or CT Enterotoxicity

The intracellular cGMP levels in T-84 cells incubated with STa toxin pretreated with the serum sample of the immunized mice were 120 ± 19.7 nM. These cGMP levels were lower than those in the cells incubated with the STa toxin pretreated with the control mouse serum samples (176 ± 65.2 nM) but were not significantly lower than the cGMP levels of 140 ± 32 nM in the cells treated with the STa toxin alone (positive control of STa toxin enterotoxicity). The baseline intracellular cGMP concentrations in T-84 cells were 5 ± 0.1 nM.

The intracellular cAMP concentrations in the T-84 cells incubated with CT toxin pretreated with the sera of the immunized mice were over 200 nM, the same as the cAMP in the cells treated with the CT toxin alone or CT toxin pretreated with the control mouse sera. The baseline cAMP levels in T-84 cells (no toxin, no sera) were 14.5 ± 7.5 nM.

### 3.5. Adhesin MEFA-IIb Protein Induced Antigen-Specific Antibody Responses in the IM-Immunized Rabbits

Rabbits IM immunized with adhesin MEFA-IIb protein developed robust IgG responses to the target adhesin and toxin antigens. Anti-CS7, -CS12, -CS14, -CS17, and anti-CS21 IgG titers were 3.4 ± 0.20, 3.7 ± 0.23, 3.9 ± 0.25, 4.0 ± 0.37 and 4.1 ± 0.18 (log_10_), respectively, from the serum samples of the immunized rabbits. Anti-LT and anti-STa IgG were detected at 1.7 ± 0.18 and 2.6 ± 0.54 (log_10_) from the immunized rabbit sera. No antigen-specific IgG antibodies were detected from the control rabbit serum samples (Figure 4). No anti-adhesin or antitoxin IgA was detected, nor was IgG detected from rabbit sera that were collected prior to the primary immunization.

Mild IgA responses to the target adhesins were detected from the cecum content of the immunized rabbits. Anti-CS7, -CS12, -CS14, -CS17, and anti-CS21 IgA titers were detected at 1.1 ± 0.31, 1.4 ± 0.59, 1.7 ± 0.64, 1.1 ± 0.11, and 1.2 ± 0.24 (log_2_). Anti-LT IgA responses were not detected in the cecum contents of the immunized rabbits, and low anti-STa IgA was detected (0.95; log_2_) from the cecum content of one rabbit. No antigen-specific IgA responses were detected from the control rabbit cecum contents.

### 3.6. Adhesin MEFA-IIb-Induced Antibodies in Rabbit Serum Samples Broadly Inhibited Adherence by ETEC Bacteria Expressing CS7, CS12, CS14, CS17, or CS21 Adhesin

After incubation with the sera of the rabbits immunized with adhesin MEFA-IIb, ETEC isolates JF2327 (CS7), JF3276 (CS12), JF2125 (CS14), JF2350 (CS17), and JF2101 (CS21) in vitro adherence to Caco-2 cells were 46%, 48%, 50%, 42%, and 56%, respectively, compared to the adherence of the same bacteria (100%) incubated with the control rabbit sera (Figure 5).

### 3.7. Rabbits IM Immunized with Adhesin MEFA-IIb Exhibited a Significant Reduction in CS21 ETEC Intestinal Colonization

Rabbits IM immunized with adhesin MEFA-IIb protein showed a significant reduction in bacteria colonization in small intestines, compared to the control rabbits after orogastric inoculation with an ETEC strain producing CS21 adhesin (JF2101) (Figure 6). ETEC bacteria recovered from the ileal distal segment from the immunized rabbits were (5.9 ± 1.5) × 10^6^ CFUs per gram tissue, which was significantly fewer than the control rabbits ((1.6 ± 1.2) × 10^7^ CFUs; *p* < 0.001). Necropsy revealed that the immunized rabbits had formed fecal pellets in the small intestine. In contrast, the control rabbits, although they did not develop diarrhea, had unformed pasty feces or yellowish fluid accumulation in the small intestine and cecum.

Ten colonies randomly picked from each plate were confirmed positive for CS21 adhesin in PCRs with primers specific to CS21 major subunit LngA.

## 4. Discussion

The results from this study show that the polyvalent ETEC adhesin MEFA-IIb protein is broadly immunogenic. When administered intramuscularly, adhesin MEFA-IIb induced robust antibody responses to the five targeted ETEC adhesins (CS7, CS12, CS14, CS17, and CS21) and both ETEC toxins (STa and LT) in mice and rabbits. If this adhesin MEFA-IIb induces functional antibodies against the five adhesins and two toxins, it can potentially be combined with another ETEC adhesin polyvalent protein, CFA/I/II/IV MEFA, which protects against the seven first-tiered ETEC adhesins (CFA/I and CS1-CS6), for a broad-spectrum ETEC subunit vaccine. Such a vaccine would protect against ETEC strains expressing any of these twelve ETEC adhesins and/or both toxins, the strains causing a vast majority of ETEC-associated diarrhea as well as moderate-to-severe clinical cases. This multiantigen vaccine candidate can be very effective against ETEC children’s diarrhea and travelers’ diarrhea.

However, while the anti-adhesin antibodies induced by adhesin MEFA-IIb were functional in the inhibition of the adherence of ETEC isolates expressing any of the five targeted adhesins and led to a significant reduction in CS21 ETEC bacteria intestinal colonization in rabbits, the protein-induced antitoxin antibodies unexpectedly did not neutralize STa or CT enterotoxicity. This rejects the idea that a combination of two polyvalent adhesin proteins, namely adhesin MEFA-IIb and CFA/I/II/IV MEFA, leads to the development of a broadly protective ETEC subunit vaccine. Instead, adhesin MEFA-IIb or adhesin MEFA-II [17] needs to be combined with MecVax [19], an ETEC subunit vaccine candidate consisting of CFA/I/II/IV MEFA and toxoid protein 3xSTa_N12S_-mnLT_R192G/L211A_. MecVax protects against the seven most important ETEC adhesins (CFA/I and CS1–CS6) and both ETEC toxins (STa and LT) [19,21,25], a combination of three polyvalent proteins, for a new subunit vaccine candidate to protect against the ETEC strains expressing any of the twelve prevalent and virulent ETEC adhesins and one or both ETEC toxins. Further studies on the examination of the antigenic compatibility of these three proteins and especially preclinical studies for broad protection are needed to assess the feasibility of combining the three proteins into an ETEC subunit vaccine product.

STa toxoid STa_N12S_ was found to induce neutralizing antibodies when presented by a carrier protein or on a backbone protein. After being genetically fused to a monomeric double mutant LT (mnLT_R192G/L211A_ or monomeric dmLT), toxoid STa_N12S_ elicited anti-STa antibodies not only neutralizing STa toxin enterotoxicity [24,25,26,27,28] but also protecting against STa toxin-mediated clinical diarrhea in pigs [19,29,30]. Similarly, when presented on chicken ovalbumin protein or ETEC CFA/I major subunit CfaB, STa_N12S_–ovalbumin fusion or adhesin MEFA-II induced neutralizing anti-STa antibodies [17,31]. By adding a second STa_N12S_ toxoid on the backbone protein CfaB, we expected adhesin MEFA-IIb to elicit stronger neutralizing anti-STa antibodies since an additional STa toxoid on a toxoid fusion protein was demonstrated to enhance STa immunogenicity and the induction of neutralizing anti-STa antibodies [32,33]. In contrast, adhesin MEFA-IIb, despite inducing anti-STa antibodies, abolished the ability to elicit neutralizing anti-STa antibodies. The precise mechanism for the loss of stimulation of neutralizing antibodies against STa toxin is currently unknown. However, unlike toxoid fusion 3xSTa_N12S_-mnLT_R192G/L211A_ and adhesin MEFA-II protein, which have STa_N12S_ positioned at the N-terminus of the protein, adhesin MEFA-IIb has two STa_N12S_ toxoids inserted at the middle of the protein (neither at the N-terminus nor the C-terminus). The positions of STa_N12S_ on the adhesin MEFA-IIb likely altered toxoid structure and thus antigenicity and further impaired the ability to induce neutralizing antibodies. Future studies on the comparative examination of the reactivity of neutralizing anti-STa antibodies with adhesin MEFA-II (induces neutralizing anti-STa antibodies) versus adhesin MEFA-IIb (does not induce neutralizing antibodies against STa toxin) may provide insight into the alteration of the STa toxoid antigenic topology.

Similarly to toxoid STa_N12S_, the epitope of the LT toxic A subunit, SPHPYEQEVSALTA induced anti-LT antibodies on the adhesin MEFA-IIb, but the derived anti-LT antibodies did not neutralize CT (LT homolog) enterotoxicity. LT_A_ epitope “SPHPYEQEVSA” was demonstrated to be strongly immunogenic and the best LT epitope in inducing neutralizing antibodies when carried by chicken ovalbumin [34]. It is unlikely that the three additional amino acid residues (at the C-terminus) would affect the epitope in inducing neutralizing anti-LT antibodies. The position of the epitope on this adhesin MEFA-IIb likely altered epitope antigenic topology, thus abolishing the ability to induce functional antibodies.

Further studies are needed to realign the STa toxoids and the LT epitope on adhesin MEFA-IIb and optimize this protein immunogen for neutralizing antitoxin antibodies. This study highlights the fact that, while immunogenic B-cell epitopes can be predicted reliably in silico, epitope functional activity can only be determined empirically. This study also suggests that a different carrier, backbone, or structural position, can affect or abolish the functionality of an otherwise neutralizing epitope.

It was observed that mice intramuscularly immunized with the polyvalent protein developed systemic IgG responses, whereas rabbits IM immunized with the same protein antigen developed systemic IgG but also a low level of seIgA (anti-adhesin IgA responses from cecum contents). We believe that this disparity is likely caused by the amount of dmLT adjuvant used in intramuscular immunization. Specifically, 0.1 μg dmLT was used as the adjuvant in mouse immunization, whereas 1 μg dmLT was included in rabbit immunization. An increasing dose of dmLT (or mLT) adjuvant and a different animal species were linked to the induction of low levels of systemic IgA responses after parenteral immunization with an ETEC protein antigen [35,36,37,38]. Future mouse intramuscular immunization with higher dmLT adjuvant doses, or with the selection of a strong mucosal adjuvant, should help us to determine if systemic IgA responses can also be elicited by the IM-immunized adhesin MEFA-IIb protein.

## 5. Conclusions

In conclusion, according to the data from this study, adhesin MEFA-IIb elicited broad antibody responses to the five targeted ETEC adhesins (CS7, CS12, CS14, CS17, and CS21) and both ETEC toxins (STa and LT). While the anti-adhesin antibodies were functional against ETEC bacteria adherence and intestinal colonization, this MEFA-IIb protein needs to be optimized to induce neutralizing antitoxin antibodies. Nevertheless, adhesin MEFA-IIb or adhesin MEFA-II [17] can potently be an add-on protein antigen to MecVax for the development of a truly broadly protective ETEC vaccine against children’s diarrhea and travelers’ diarrhea.

## Figures and Tables

**Figure 1 microorganisms-11-02473-f001:**
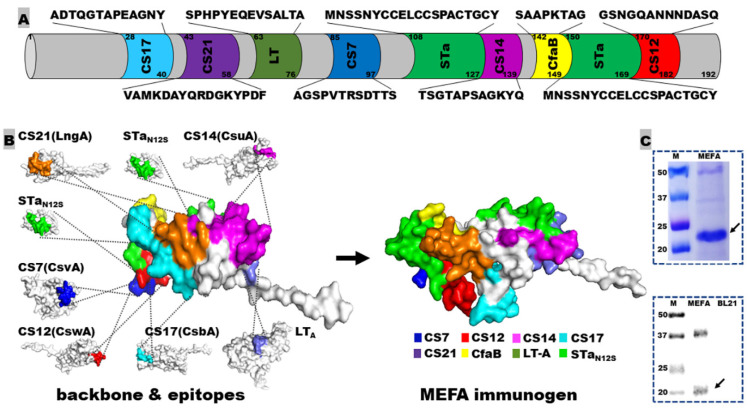
Adhesin MEFA-IIb protein structure illustration and characterization: (**A**) a scheme to show epitope sequence and genetic position on adhesin MEFA-IIb; (**B**) construction of the adhesin MEFA-IIb protein and protein structure model, with each epitope colored differently; (**C**) visualization of the adhesin MEFA-IIb protein in SDS-PAGE Coomassie blue staining and detection in Western blot with anti-CS21 mouse polyclonal antibodies.

**Figure 2 microorganisms-11-02473-f002:**
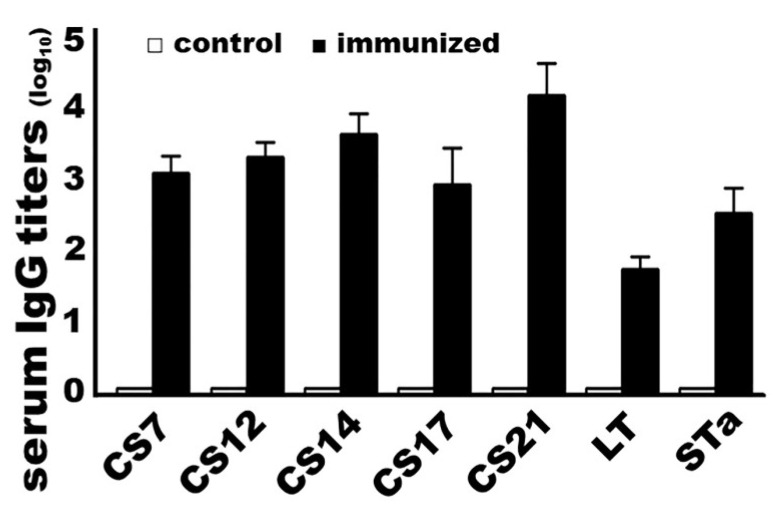
Mouse serum anti-adhesin and antitoxin IgG titers (log_10_) from the group intramuscularly immunized with adhesin MEFA-IIb (black boxes) or PBS (white boxes; as the negative control). Recombinant major subunit of each adhesin, cholera toxin (CT; an LT homolog), and STa–ovalbumin conjugates were used as ELISA coating antigens. HRP-conjugated goat anti-mouse IgG (1:5000) was used as the secondary antibody. Boxes and bars indicate the mean titers and standard deviations.

**Figure 3 microorganisms-11-02473-f003:**
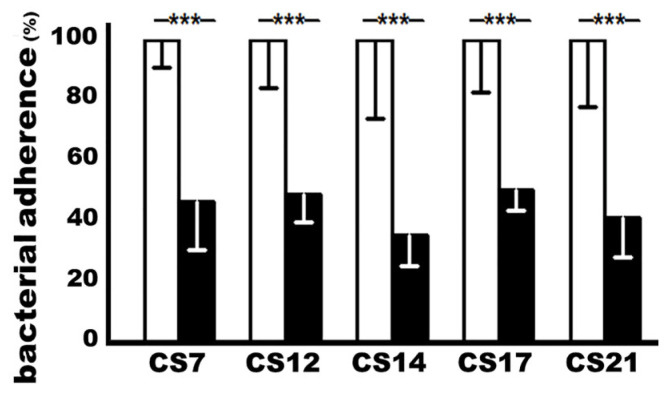
The CS7, CS12, CS14, CS17, or CS21 ETEC bacteria adherence to Caco-2 cells after treatment with the sera of the immunized or control mice; antibody adherence inhibition assay was used to show antibody in vitro protection against bacterial adherence. Each ETEC strain, pretreated with 10% mannose, was incubated with the heat-inactivated sera from the immunized mice (black boxes) or the control mice (white boxes) and applied to the Caco-2 cells. The adherent bacteria were collected and cultured on LB plates, and bacteria were counted (in CFUs, colony-forming units) after overnight growth. The CFUs from the treatment with the control mouse sera were referred to as 100%. Boxes and bars indicate means and standard deviations, and *** a *p*-value of less than 0.001.

**Figure 4 microorganisms-11-02473-f004:**
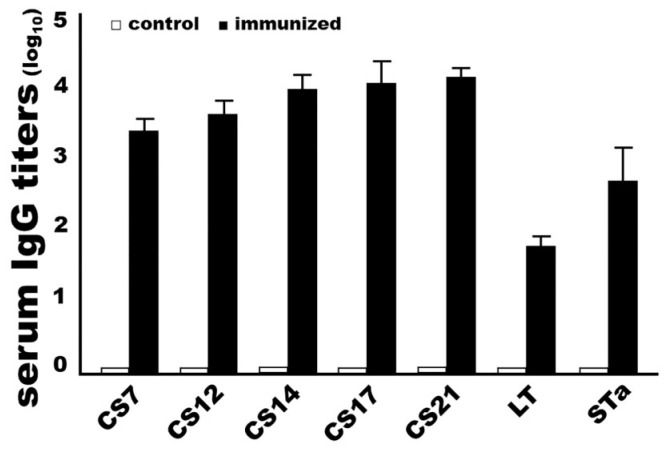
Antigen-specific IgG titers (log10) detected from the sera of the rabbits intramuscularly immunized with adhesin MEFA-IIb (black boxes) or PBS (white boxes). Recombinant major subunit of each adhesin, cholera toxin (CT), and STa–ovalbumin conjugates were used as the coating antigens. HRP-conjugated goat anti-rabbit IgG (1:5000) was used as the secondary antibody. Boxes and bars indicate the mean titers and standard deviations.

**Figure 5 microorganisms-11-02473-f005:**
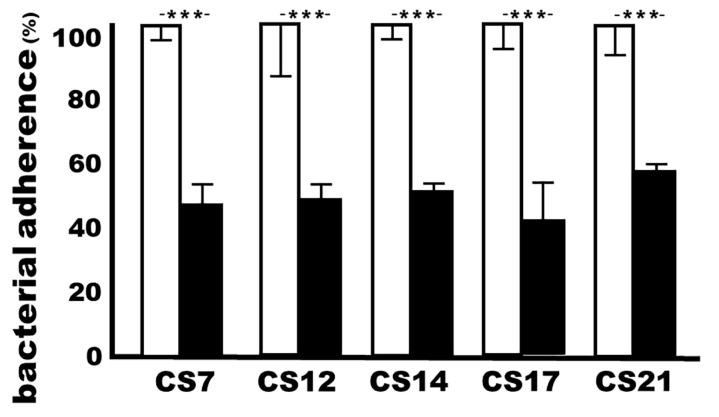
The CS7, CS12, CS14, CS17, or CS21 ETEC adherence (%) to Caco-2 cells after treatment with the sera of the rabbits IM immunized with adhesin MEFA-IIb. ETEC bacteria, pretreated with 10% mannose, were incubated with heat-inactivated sera from the immunized rabbits (black boxes) or control rabbits (white boxes) and then transferred to the Caco-2 cells. The adherent bacteria (CFUs) were counted, and CFUs from the treatment with control rabbit sera were considered to be 100%. Boxes and bars indicate means and standard deviations, and *** a *p*-value of less than 0.001.

**Figure 6 microorganisms-11-02473-f006:**
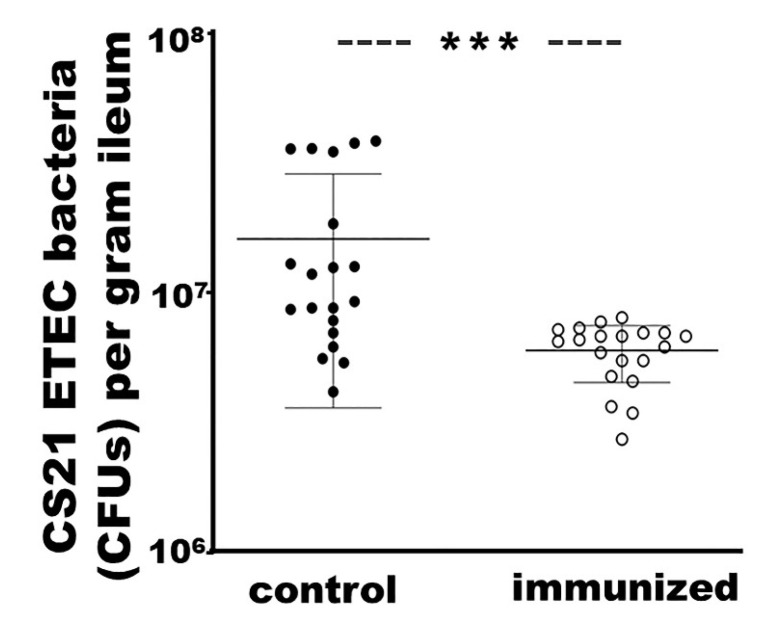
Intramuscular immunization with MEFA-IIb protein protected against ETEC isolate JF2101 (CS21 and LT) colonization in rabbit small intestines. NZW adult rabbits IM immunized with MEFA-IIb (empty circle) or PBS (solid circle) were orogastrically inoculated with JF2101 inoculum, and after 24 h post-inoculation, rabbits were euthanized; an ileal distal segment was collected from each rabbit, ground, serially diluted, and plated on agar plates. CFUs were counted after overnight growth. Bars indicate CFU means and standard deviations, and *** a *p*-value of < 0.001.

## Data Availability

Data are fully described in this manuscript.

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
