# Peer review of "A Polyvalent Adhesin–Toxoid Multiepitope-Fusion-Antigen-Induced Functional Antibodies against Five Enterotoxigenic Escherichia coli Adhesins (CS7, CS12, CS14, CS17, and CS21) but Not Enterotoxins (LT and STa)"

_microorganisms, 2023, doi:10.3390/microorganisms11102473_

Round 1
Reviewer 1 Report
The manuscript is an interesting development study of a vaccine using an epitope- and structure-based multiepitope-fusion-antigen (MEFA) vaccinology platform. The researchers (authors of the article) have experience, having published previous articles on the same subject. The study reports a new antigenic construct that was designed in silico. The methodological procedures for synthesizing the multiepitope antigen and evaluating the immunogenicity are adequate and well written. The results are presented in appropriate figures and graphs and the Discussion is relatively well elaborated. My opinion is that the article has scientific and technological relevance, therefore it deserves to be published.
However, I would like to suggest some modifications as well as to recommend corrections to the authors:
1) Please avoid using abbreviations in the main title and Results topics. I even strongly recommend removing all abbreviations from the title (CS7, CS12, CS14, CS17, CS21, LT, STa). I think it could be: "A polyvalent adhesin-toxoid multiepitope-fusion-antigen (MEFA-IIb) induced functional antibodies against five enterotoxigenic Escherichia coli proteins”. In the Results topics, please summarize all them.
2) What does WASH mean on the line 38? Please inform.
3) I missed some epidemiological information of ETEC in the Introduction. I think a brief description could be included in the first paragraph (starting on line 4).
4) The Discussion could be improved and deepened by comparing with more studies that developed multiple epitope vaccines.
Very good!
Author Response
Point-by-point response to reviewer’s comments
- Please avoid using abbreviations in the main title and Results topics. I even strongly recommend removing all abbreviations from the title (CS7, CS12, CS14, CS17, CS21, LT, STa). I think it could be: "A polyvalent adhesin-toxoid multiepitope-fusion-antigen (MEFA-IIb) induced functional antibodies against five enterotoxigenic Escherichia coli proteins”. In the Results topics, please summarize all them.
Response: We appreciate the comment to shorten the title. We removed the abbreviation of MEFA, but retained the CFA and toxin names for specificity, particularly there are over 25 difference adhesins produced by different ETEC strains.
2) What does WASH mean on the line 38? Please inform.
Response: Water, sanitation, and hygiene, were included in the revision.
3) I missed some epidemiological information of ETEC in the Introduction. I think a brief description could be included in the first paragraph (starting on line 4).
Response: Two reasons we did not include ETEC epidemiology or disease burden in the original submission. First, a few studies done recently, such as FERG (foodborne disease burden epidemiology reference group), GBD (global burden of disease), and CHERG/MCEE (Child health epidemiology reference group/maternal child epidemiology estimation), published detailed information of ETEC epidemiology. The second reason is that these systemic studies provided quite variable figures regarding ETEC epidemiology particularly of the annual death rate. That leaves us (other researchers as well) unclear that which dataset we can objectively cited. Nevertheless, we added a brief statement sentence (ETEC strains are estimated to cause hundreds of millions of diarrhea clinical cases and up to nearly a hundred thousand deaths annually) in the revision.
4) The Discussion could be improved and deepened by comparing with more studies that developed multiple epitope vaccines.
Response: Additional discussion was added to the revision.

Reviewer 2 Report
Dear authors,
congratulations to your work and results! ETEC-caused diarrhea causes in developing countries significant burden in childhood population and still there is no solution on table. Your methods are appropriate and results excellent. Adhesin MEFA-IIb induced antibody responses to the five targeted ETEC adhesins (CS7, CS12, CS14, CS17, CS21) and 464 both ETEC toxins (STa, LT) which are key virulence factors in children’s diarrhea and travelers’ diarrhea. Therefore adhesin MEFA-IIb, or the adhesin MEFA-II could be potential molecules used in human vaccine.
Author Response
We greatly appreciate the comment and encouragement from this reviewer. Indeed, another CFA antigen to cover the other five important ETEC adhesins can potentially expand the efficacy of our current protein-based ETEC vaccine candidate (MecVax)